# Composition Characterization of *Cinnamomum osmophloeum* Kanehira Hydrosol and Its Enhanced Effects on Erectile Function

**DOI:** 10.3390/plants13111518

**Published:** 2024-05-31

**Authors:** Chung-Hsuan Wang, Nai-Wen Taso, Chi-Jung Chen, Hung-Yi Chang, Sheng-Yang Wang

**Affiliations:** 1Special Crop and Metabolome Discipline Cluster, Academy Circle Economy, National Chung Hsing University, Taichung 402202, Taiwan; piscium0312@smail.nchu.edu.tw (C.-H.W.); nwt1228@dragon.nchu.edu.tw (N.-W.T.); 2Department of Forestry, National Chung Hsing University, Taichung 402202, Taiwan; 3Taichung Branch, Forestry and Nature Conservation Agency, Ministry of Agriculture, Taichung 402212, Taiwan; chenchijung@gmail.com (C.-J.C.); changgingo@gmail.com (H.-Y.C.); 4Agricultural Biotechnology Research Center, Academia Sinica, Taipei City 115201, Taiwan

**Keywords:** *Cinnamomum osmophloeum* Kanehira, hydrosol, PDE5, erectile dysfunction

## Abstract

*Cinnamomum osmophloeum* Kanehira (CO) is an endemic species of Taiwan. This study elucidated the composition of CO hydrosol, revealing *trans*-cinnamaldehyde (65.03%), *trans*-cinnamyl acetate (7.57%), and coumarin (4.31%) as the main volatile compounds. Seven compounds were identified in the water fraction of hydrosol, including a novel compound, 2-(2-hydroxyphenyl)oxetan-3-ol. This marks the first investigation into high-polarity compounds in hydrosol, extending beyond the volatile components. Notably, two compounds, *trans*-phenyloxetan-3-ol and *cis*-phenyloxetan-3-ol, demonstrated significant inhibition activity against phosphodiesterase type five (PDE5), with IC50 values of 4.37 µM and 3.40 µM, respectively, indicating their potential as novel PDE5 inhibitors. Furthermore, CO hydrosol was evaluated against enzymes associated with erectile dysfunction, namely acetylcholinesterase (AChE), angiotensin-I converting enzyme (ACE), and arginase type 2 (ARG2). These findings underscore the potential of CO hydrosol to modulate erectile function through diverse physiological pathways, hinting at its prospects for future development in a beverage or additive with enhanced effects on erectile function.

## 1. Introduction

Hydrosol, also known as floral waters, is the water obtained by the condensation of plants in the process of water distillation or steam distillation to extract essential oils. Although hydrosol and essential oils are not miscible, they often share similar properties, such as fragrance, and contain water-soluble ingredients. Terms like herbal distillates, floral waters, hydrolates, herbal water, and essential waters are often used interchangeably with hydrosol in the market and articles [1]. Hydrosols have various uses in natural fragrances, lotions, creams, facial toners, and other skin care products or toiletries. They can be added to bath water as eau de toilette, cologne, body spray, or even wet wipes for fragrance. According to Market Research Future, the global hydrosol market is estimated to grow at a rate of 5.17% from 2019 to 2024, and the market value is expected to reach USD 437 million by 2024. Europe currently dominates the hydrosol market, accounting for 39.9% of the global market share in 2018, mainly due to France and Italy’s reputation as distribution centers of world-renowned perfumes and fragrances, along with the rising demand for natural and organic ingredients in personal care. The European market has been consistently increasing and has become the central hub of hydrosol trade [2].

Hydrosols have a composition that is similar to essential oils, but with fewer components, as noted in various studies [3,4,5,6,7,8,9]. Hydrosols are mainly utilized for their antioxidative and antibacterial properties [7,9,10,11] and are often used for food or fruit preservation [12,13,14]. Despite the widespread use of hydrosols, there is a lack of scientific evidence to support their effectiveness. Some studies have even suggested that hydrosols are less biologically active than essential oils [15,16,17]. The main reason for this is that the chemical composition of hydrosol is still unclear. Currently, the analysis of hydrosol components is primarily focused on volatile compounds, while research on non-volatile components, such as polar components, is lacking.

Cinnamon is a highly valued spice in both Eastern and Western cultures. It has a long history of use in traditional oriental medicine as a medicinal material. *Cinnamomum osmophloeum* Kanehira (CO) is a tree species endemic to Taiwan, and its leaf essential oil has a composition similar to that of cinnamon. The CO leaf methanolic extract is highly sweet, and the compound responsible for this sweetness has been identified as *trans*-cinnamaldehyde, which is present at 1.03% *w*/*w* [18]. The essential oils extracted from CO leaves have been shown to possess bioactivity against various organisms, including bacteria [19], termites [20], mildew [21], and fungi [22]. Additionally, we have investigated the potential use of CO leaves in food supplements and found that the essential oils and their dominant compound, *trans*-cinnamaldehyde, exhibit potent xanthine oxidase inhibitory activity and have an anti-hyperuricemia effect in mice [23]. Apart from essential oils, oral administration of CO hot water extracts or leaves to hyperlipidemic hamsters has been found to reduce total cholesterol, triglyceride, and low-density lipoprotein cholesterol levels [24,25]. More recently, our research demonstrated that CO effectively lowers hyperglycemia and improves metabolic syndrome in obese mice. We also observed that CO improved gut microbiota dysbiosis by reducing the Firmicutes-to-Bacteroidetes ratio and increasing the abundance of *Akkermansia* spp. These findings suggest that CO has the potential to serve as a prebiotic dietary supplement to alleviate obesity-related metabolic disorders and gut dysbiosis [26].

Compared to essential oils, research on the activity and components of hydrosol is relatively scarce. In Taiwan, CO is a special and essential spicy plant, and its hydrosol produced during the essential oil production process has found widespread use. However, there are barely any scientific studies on the compositional analysis and activity exploration of CO hydrosol. Therefore, this study aimed to analyze the components of CO hydrosol and investigate its inhibitory activity on PDE5. PDE5 inhibitors have proven effective in treating erectile dysfunction by relaxing smooth muscle and increasing cGMP levels in smooth muscle cells, leading to the production of more endothelial nitric oxide synthase (eNOS) and nitric oxide (NO). Our research has discovered that CO hydrosol and its components significantly inhibit the activity of the PDE5 enzyme.

## 2. Results and Discussion

### 2.1. Chemical Composition of CO Essential Oil and Hydrosol

The major constituents of CO essential oil and the hydrosol EA fraction were determined via GC-MS analysis, and their relative contents (%) are presented in Table 1. In the essential oil, the prominent compounds include *trans*-cinnamaldehyde (68.38%), *trans*-cinnamyl acetate (18.52%), and *β*-caryophyllene (1.87%). Conversely, the hydrosol EA fraction contains *trans*-cinnamaldehyde (65.03%), *trans*-cinnamyl acetate (7.57%), and coumarin (4.31%). The GC-MS analysis revealed that the chemical composition was similar between CO essential oil and the EA fraction.

The lyophilization hydrosol water fraction underwent C-18 silica gel column chromatography, yielded three sub-fractions (A–C). Sub-fraction A constituted 49.3% of the total water fraction and was purified via reverse-phase HPLC (Figure 1). It was then analyzed by NMR and mass spectrometry (MS), resulting in the purification of seven compounds (The NMR spectra were shown in the Appendix A).

Based on the NMR spectr0 and compared with the literature, compounds **1**–**6** were identified as *trans*-2-methyloxetan-3-ol (**1**) [27], *cis*-2-methyloxetan-3-ol (**2**) [27], 3-methoxy-4-hydroxyphenylglycol (**3**) [28], *cis*-phenyloxetan-3-ol (**4**) [29,30], 5-(2-hydroxypropan-2-yl)-2-methylcyclohex-3-ene-1,2-diol (**5**) [31], and *trans*-phenyloxetan-3-ol (**6**) [29,30] (Figure 2). Compounds **1**, **2**, **4**, and **6** were previously reported as synthetic intermediates, and this is the first time they were obtained from natural products. Interestingly, compound **5** was isolated from lavender hydrosol [31], and we identified the same component in different hydrosols.

Compound **7** showed a molecular formula of C_9_H_10_O_3_ established by HRMS ([M − H]^−^ *m*/*z* 165.0559). The ^1^H NMR spectrum displayed a methylene proton at δ_H_ 4.15 (1H, ddd, *J* = 11.6, 4.8, 1.2 Hz; 1H, dd, *J* = 11.6, 2.4 Hz); two methine protons at δ_H_ 3.88 (1H, td, *J* = 4.4, 2.4 Hz) and δ_H_ 4.47 (1H, d, *J* = 4.4 Hz); and four aromatic protons, showing a ortho substitution benzene at δ_H_ 6.81 (1H, dd, *J* = 8.4, 0.8 Hz), δ_H_ 6.93 (1H, td, *J* = 7.6, 0.8 Hz), δ_H_ 7.18 (1H, td, *J* = 8.4, 1.6 Hz), and δ_H_ 7.36 (1H, dd, *J* = 7.6, 1.6 Hz). The methylene and methine protons exhibited downfield chemical shifts, indicating their attachment to the oxygen and inducing deshielding effects.

The ^13^C, DEPT, and HSQC experiments displayed nine carbon signals, including two methine carbons at δ_C_ 69.1 (C-2) and 69.3 (C-3); a methylene carbon at δ_C_ 67.2 (C-4); and an aromatic system at δ_C_ 124.2 (C-5), 155.5 (C-6), 117.5 (C-7), 130.3 (C-8), 121.8 (C-9), and 132.0 (C-10). C-2, C-3, C-4, and C-6 were linked to oxygen. The degrees of unsaturation were 5, including an aromatic system with no indication of an olefinic bond, suggesting the presence of another ring system in the structure. The HMBC experiment showed H-4a/C-3, and C-6 indicated that the ring system was a lactone structure. Based on the above evidence, compound **7** was elucidated as 2-(2-hydroxyphenyl)oxetan-3-ol (Figure 2). Table 2 presents the ^1^H and ^13^C NMR chemical shift data of compound **7**.

### 2.2. PDE5 Inhibition Activity of Different CO Extracts

Phosphodiesterase type V (PDE5) is a cGMP-specific hydrolase. The inhibition of PDE5 decreases the breakdown of cGMP and causes vasodilation in the penis and lungs. Different types of CO extracts were used for the in vitro PDE5 inhibition assay, including methanol extracts, hot water extracts, essential oil, and hydrosol. Additionally, 10 nM sildenafil was used as a positive control [32]. As shown in Figure 3, except for the essential treatment, all other extracts revealed good PDE5 inhibition activity, especially the hydrosol treatment, which demonstrated efficacy at low concentrations. The PDE5 inhibition rates at the dosages of 0.025, 0.05, and 0.1 µg/mL of hydrosol were 35.4%, 49.9%, and 70.7%, respectively.

Through component analysis, it could be deduced that the chemical composition of CO hydrosol closely resembles that of essential oil. Notably, the ethyl acetate fraction constitutes 0.13% of the hydrosol, with trans-cinnamaldehyde emerging as the principal component. This suggests the presence of potentially potent compounds within the water fraction, which constitutes a mere 0.044% of the hydrosol, exhibiting notable inhibitory effects on PDE5. Consequently, additionally activity assays and component analysis were carried out on the hydrosol water fraction.

### 2.3. PDE5 Inhibition Activity of CO Hydrosol and Its Active Compounds

As shown in Figure 4, the hydrosol ethyl acetate fraction presented a similar PDE5 inhibition effect to essential oil. Their inhibition rates at the dosages of 100 µg/mL were 25.5% and 25.8%, respectively. This was expected as they contain similar chemical compositions. The main component in both is trans-cinnamaldehyde. In Dell’ Agli’s study, the inhibition rate of *trans*-cinnamaldehyde at a concentration of 10 µM was less than 20% [33], which also corroborates our findings. The PDE5 inhibition rates at the dosages of 1.25, 2.5, 5, 10, and 20 µg/mL of the hydrosol water fraction were 7.4%, 31.4%, 36.9%, 42.4%, and 47.5%, respectively. The treatment of the hydrosol water fraction with increasing concentrations significantly as well as dose-dependently reduced the PDE5 activity. The results indicated that the components in CO hydrosol responsible for PDE5 inhibition were primarily distributed in the water fraction. Therefore, subsequent chromatographic and spectroscopic techniques were employed to isolate and identify (seven) compounds from the CO hydrosol water fraction, followed by activity screening.

When conducting PDE5 inhibition assays on the seven isolated compounds, *trans*-phenyloxetan-3-ol and *cis*-phenyloxetan-3-ol exhibited significant inhibition activity, with IC_50_ values of 4.37 µM and 3.40 µM, respectively. The IC_50_ values of the remaining five compounds were all higher than 40 µM. The inhibition rates of *trans*-phenyloxetan-3-ol at concentrations of 0.1, 0.5, 1, 10, and 20 µM were 11.5%, 26.1%, 36.7%, 58.4%, and 65.3%, respectively, and those of *cis*-phenyloxetan-3-ol at concentrations of 0.1, 0.5, 1, 5, and 10 µM were 10.8%, 27.1%, 38.1%, 52.1%, and 61.9%, respectively, indicating that *cis*-phenyloxetan-3-ol exhibits better inhibition of PDE5 activity (Figure 5). Epimedium, known as a traditional herbal remedy for male enhancement, contains the active compound icariin, which has been isolated and found to exhibit good inhibitory effects against PDE5, with an IC_50_ value of approximately 5.9 µM [33]. However, the two active compounds we found in CO hydrosol were both shown to have lower IC_50_ values than icariin. They may be the most potent natural compounds published to date for PDE5 inhibition.

### 2.4. ACE Inhibition Activity of CO Hydrosol

In vivo, angiotensin I is converted by ACE into the physiologically active angiotensin II, which raises the blood pressure and induces vasoconstriction. Vasoconstriction not only impacts the erectile process but also contributes to long-term vascular wall damage due to chronic hypertension [34]. As shown in Figure 6, both of the EA fraction and water fraction of hydrosol presented a dose-dependent inhibition effect. The EA fraction exhibited better inhibitory effects than the water fraction, and their IC_50_ values were 966.7 µg/mL and 1630.0 µg/mL. The results indicated that CO hydrosol exhibited a weaker inhibitory effect on ACE.

### 2.5. AChE Inhibition Activity of CO Hydrosol

AChE inhibitors can be used clinically as a treatment for Alzheimer’s disease because they can increase the concentration of acetylcholine in the brain, thereby enhancing cholinergic neurotransmission [35]. Acetylcholine can also activate eNOS in endothelial cells, influencing erectile function. As shown in Figure 7, the EA fraction exhibited AChE inhibition rates of 9.1%, 27.1%, 57.5%, 68.8%, and 75.3% at concentrations of 400, 800, 1600, 2000, and 2400 µg/mL, respectively. The water fraction showed inhibition rates of 15.8%, 27.2%, 45.9%, 52.7%, and 62.0% at concentrations of 400, 800, 1600, 2000, and 2400 µg/mL, respectively. Their IC_50_ values were 1342.8 µg/mL and 1671.1 µg/mL. The results indicated that the EA fraction presented a better inhibition effect than the water fraction.

### 2.6. ARG2 Inhibition Activity of CO Hydrosol

Arginase is an enzyme of the urea cycle that catalyzes the hydrolysis of _L_-arginine to _L_-ornithine and urea. Two isoforms coexist, with type I predominantly expressed in the liver and the type II arginase expressed throughout extrahepatic tissues [36]. Circulating ARG2 concentrations increase in clinical ED, associated with an increased risk of ED [37]. As shown in Figure 8, the EA fraction exhibited ARG2 inhibition rates of 31.3%, 39.2%, 50.2%, 61.4%, and 71.4% at concentrations of 100, 200, 400, 800, and 1600 µg/mL, respectively. The water fraction showed inhibition rates of 26.3%, 35.4%, 47.3%, 55.7%, 66.8%, and 73.1% at concentrations of 100, 200, 400, 800, 1600, and 2400 µg/mL, respectively. Their IC_50_ values were 382.0 µg/mL and 513.6 µg/mL. The inhibition effect of ARG2 is greater than that of ACE and AChE, and the EA fraction exhibited better activity than the water fraction in all enzymes except for PDE5.

## 3. Materials and Methods

### 3.1. General Experimental Procedures

NMR experiments (^1^H, ^13^C, HSQC, HMBC, and COSY) were carried out on a Bruker AVANCEIII 400 spectrometer (Bruker, Billerica, MA, USA) at 300 K using CD_3_OD or D_2_O as the solvent. ESI-MS were obtained in the positive and negative ion modes on a Bruker amaZon speed mass spectrometer (Bruker). Column chromatography was performed over RP-18 silica gel (40–63 µm; Merck, Darmstadt, Germany). RP-HPLC separation was conducted on an Agilent 1100 series system equipped with a COSMOSIL C18-AR-II (4.6 mm I.D. × 250 mm, Nacalai Tesque Inc., San Diego, CA, USA), which was employed with a MeOH/H_2_O solvent system. The mobile phase condition is as shown in Table 3.

### 3.2. Plant Material

The leaves for essential oil and hydrosol extraction were collected from a batch of 15-year-old CO trees in June 2021 from Chuyunshan Nursery, Taichung City, Taiwan, and the species was identified by Prof Sheng-Yang Wang, Department of Forestry, National Chung Hsing University. A voucher specimen (C.J. Chen s. n., TCF) was deposited in the herbarium of the same university. The leaves were washed and air-dried, then stored at −20 °C until used.

### 3.3. Preparation of CO Leaf Essential Oil and Hydrosol

The hydrosol of CO leaves was prepared using the steam distillation method. Approximately 30 kg of CO leaves was distilled for 8 h distillation time. Following distillation, the steam containing essential oils and hydrosols was condensed and collected separately. The yield of CO essential oil was 3.06 mL/kg. Experiments were conducted using the hydrosol collected from the initial 20 L. Upon partitioning with ethyl acetate, we obtained 25.6 g of the EA fraction, with a yield of 0.13% (*w*/*v*). The hydrosol water fraction was lyophilized and yielded 885.6 mg, with a yield of 0.0044% (*w*/*v*).

### 3.4. Compounds of CO Hydrosol Water Fraction Identification 

The lyophilization of the CO hydrosol water fraction (885.6 mg) was subject to RP-18 silica gel column chromatography by eluting with water followed by three concentrations of methanol (30%, 60%, and 100%), which yielded three sub-fractions (A–C). The weights of three sub-fractions were 436.8 mg, 174.6 mg, and 101.4 mg, respectively. The A sub-fraction accounts for 49.3% of the total water fraction; therefore, it was analyzed by RP- HPLC, and seven compounds (**A1**–**A7**) were purified.

*trans*-2-methyloxetan-3-ol (**1**) ESIMS *m*/*z* 89.6 [M + H]^+^; ^1^H NMR (in D_2_O): δ (ppm) 1.34 (3H, d, *J* = 6.8, 3 Hz), 3.73 (1H, dd, *J* = 12.0, 5.6 Hz), 3.82 (1H, dd, *J* = 12.0, 5.6 Hz), 4.11 (1H, m), 4.14 (1H, m).

*cis*-2-methyloxetan-3-ol (**2**): ESIMS *m*/*z* 89.1 [M + H]^+^; ^1^H NMR (CD_3_OD): δ (ppm) 1.18 (3H, d, *J* = 6.8, 3 Hz), 3.40 (1H, m), 3.53 (1H, dd, *J* = 11.2, 6.4 Hz), 3.64 (1H, dd, *J* = 11.2, 4.8 Hz), 3.75 (1H, m).

3-methoxy-4-hydroxyphenylglycol (**3**): ESIMS *m*/*z* 206.9 [M + Na]^+^; ^1^H NMR (CD_3_OD): δ (ppm) 3.59 (2H, d, *J* = 6.0 Hz), 3.86 (3H, s), 4.59 (1H, t, *J* = 6.0 Hz), 6.75 (1H, d, *J* = 8.0 Hz), 6.79 (1H, dd, *J* = 8.0, 1.6 Hz), 6.96 (1H, d, *J* = 1.6 Hz). 

*cis*-phenyloxetan-3-ol (**4**): ESIMS *m*/*z* 150.9 [M + H]^+^; ^1^H NMR (CD_3_OD): δ (ppm) 3.59 (1H, dd, *J* = 11.2, 6.8 Hz), 3.66 (1H, dd, *J* = 11.2, 3.6 Hz), 3.75 (1H, ddd, *J* = 6.8, 6.0, 3.6 Hz), 4.61 (1H, d, *J* = 6.0 Hz), 7.25 (1H, m), 7.33 (2H, m), 7.40 (2H, dd, *J* = 9.2, 2.0 Hz).

5-(2-hydroxypropan-2-yl)-2-methylcyclohex-3-ene-1,2-diol (**5**): ESIMS *m*/*z* 187.1 [M + H]^+^; ^1^H NMR (CD_3_OD): δ (ppm) 1.16 (3H, s), 1.21 (3H, s), 1.25 (3H, s), 1.83 (2H, m), 2.34 (1H, m), 3.72 (1H, dd, *J* = 6.0, 2.0 Hz), 5.58 (1H, ddd, *J* = 10.4, 2.4, 1.6 Hz), 5.88 (1H, dd, *J* = 10.4, 1.6 Hz).

*trans*-phenyloxetan-3-ol (**6**): ESIMS *m*/*z* 151.0 [M + H]^+^; ^1^H NMR (CD_3_OD): δ (ppm) 3.36 (1H, dd, *J* = 11.2, 6.4 Hz), 3.50 (1H, dd, *J* = 11.2, 4.4 Hz), 3.69 (1H, ddd, *J* = 6.4, 6.0, 4.4 Hz), 4.63 (1H, d, *J* = 6.0 Hz), 7.25 (1H, m), 7.33 (2H, m), 7.40 (2H, dd, *J* = 7.2, 2.0 Hz).

2-(2-hydroxyphenyl)oxetan-3-ol (**7**): ESIMS *m*/*z* 165.0559 [M-H]^−^; ^1^H NMR (CD_3_OD): δ (ppm) 3.88 (1H, td, *J* = 4.4, 2.4 Hz), 4.10 (1H, ddd, *J* = 11.6, 4.8, 1.2 Hz), 4.21 (1H, dd, *J* = 11.6, 2.4 Hz), 4.47 (1H, d, *J* = 4.4 Hz), 6.81 (1H, dd, *J* = 8.4, 0.8 Hz), 6.93 (1H, td, *J* = 7.6, 0.8 Hz), 7.18 (1H, td, *J* = 8.4, 1.6 Hz), 7.36 (1H, dd, *J* = 7.6, 1.6 Hz).

### 3.5. GC/MS Analysis

The chemical composition of CO essential oil or hydrosol ethyl acetate fraction was analyzed by a Thermo TRACE GC Ultra gas chromatograph (Thermo Fisher Scientific, Waltham, MA, USA) coupled with an ITQ900 mass spectrometer. An ITQ 900 mass spectrometer (Thermo Fisher Scientific, Waltham, MA, USA) was coupled with a DB-5MS column, and the temperature program was as follows: 40 °C for 3 min, then increased to 3 °C/min to 180 °C, and then increased to 20 °C/min to 280 °C, held for 5 min. The other parameters were injection temperature, 240 °C; ion source temperature, 200 °C; EI, 70 eV; carrier gas, He 1 mL/min; and mass scan range, 40–600 *m*/*z*. The volatile compounds were identified by referring to the Wiley/NBS Registry of mass spectral databases (V. 8.0, Hoboken, NJ, USA), National Institute of Standards and Technology (NIST) Ver. 2.0 GC/MS libraries, and the Kovats indices (KIs) were calculated for all volatile constituents using a homologous series of *n*-alkanes C_9_–C_24_. The major components were identified by co-injection with standards (wherever possible).

### 3.6. PDE5 Inhibition Assay

The PDE5 assay was determined by using a PDE5A1 assay kit (BPS Bioscience, San Diego, CA, USA). The experiments were conducted following the manufacturer’s protocol. Sildenafil (10 nM) was used as a positive control.

### 3.7. ACE Inhibition Assay

The ACE inhibition assay was performed via the LC-MS method as described previously with modifications [38]. Briefly, a 500 µL reaction mixture containing 400 µL of 6.25 mM hippuryl-_L_-histidyl-_L_-leucine in Tris buffer, 80 µL of rabbit lung ACE (25 U/mL), 16 µL of 50 mM Tris buffer (pH 7.4), and 4 µL of various concentration of test sample was added to a 1.5 mL Eppendorf tube and incubated on a shaker at 37 °C for 2 h. Next, 500 µL of 1M HCl was added to terminate the reaction. After filtration of the reaction mixture using a 0.22 μm membrane filter, 2 μL was injected into LC-MS/MS for the analysis of the hippuric acid content to calculate the ACE inhibition rate. The mobile phase consisted of acetonitrile (A) and water (mixed with 0.1% formic acid, B) at a flow rate of 0.3 mL/min. The initial mobile phase composition was 90% B, which was changed to 80% B at 5 min and 100% B at 7 min, then maintained for 3 min and changed to 100% A at 11 min. The UV detector was set at 254 nm. Captopril (20 nM) was used as a positive control.

### 3.8. AChE Inhibition Assay

The AChE inhibition assay was performed via the ELISA microplate reader method, which was modified from the method described by Di Giovanni [39]. Briefly, a 300 µL reaction mixture, containing 10 µL of 4.05 mM 5,5-dithio-bis-(2-nitrobenzoic acid) (DTNB), 20 µL of AChE (37 mU/mL), 227.3 µL of 100 mM phosphate buffer (pH 7.4), 2.7 µL of various concentration of test sample, and 20 µL of substrate solution consisting of 3.375 mM acetylthiocholine iodide (ATCI), was added into a 96-cell microplate. The absorbance of reaction product was measured every 20 s for 3 min at 412 nm using a micro-plate reader (Biotek Instruments, Winooski, VT, USA). Neostigmine methylsufate (1 µM) was used as a positive control.

### 3.9. Arginase 2 Inhibition Assay

The ARG2 inhibition assay was conducted using the ELISA microplate reader method as described previously with modifications [40]. Briefly, a 250 µL reaction mixture, containing 190 µL of 10 mM MnCl_2_ in 50 mM Tris-HCl buffer (pH 7.5), 50 µL of human recombinant ARG2 (25 U/mL), 100 µL of 50 mM arginine (pH 9.7), and 10 µL of various concentrations of test sample, was added to a 1.5 mL Eppendorf and incubated on a shaker at 37 °C for 1 h. Next, 600 µL of H_2_SO_4_/H_3_PO_4_/H_2_O (1:3:7) was added to terminate the reaction. Finally, 50 μL of α-isonitrosopropiophenone dissolved in alcohol (50 mg/mL) was added. The reaction mixture was then incubated at 100 °C in the dark for 45 min. After cooling, the mixture was centrifuged at 10,000× *g* for 10 min. Subsequently, the supernatant was transferred to a 96-cell microplate in the dark, and the absorbance was measured at 550 nm using a microplate reader (Biotek Instruments). *N*^ω^-hydroxy-nor-_L_-arginine (1 µM) was used as a positive control.

### 3.10. Statistical Analysis

Data were expressed as the mean ± SD of three independent experiments. Statistical analysis was performed using GraphPad Prism 9.5 for Windows (GraphPad Software, La Jolla, CA, USA). Statistical significance was scored by using one-way ANOVA followed by Tukey’s test for multiple comparison. * *p* < 0.05, ** *p* < 0.01, and **** p* < 0.001 were considered statistically significant from sample treatment groups vs. the control group.

## 4. Conclusions

In our investigation, we undertook a thorough examination of CO hydrosol, revealing a composition akin to that of essential oil, notably dominated by *trans*-cinnamaldehyde. Within the highly polar (water soluble) fraction, our investigation isolated and identified seven compounds. Among these, four (designated as compounds **1**, **2**, **4**, and **6**) were discovered for the first time from natural sources, while 2-(2-hydroxyphenyl)oxetan-3-ol (compound **7**) emerged as a newly unearthed compound. Expanding upon our discoveries, we observed that CO hydrosol demonstrated promising inhibitory activity against PDE5. Particularly noteworthy were *trans*-phenyloxetan-3-ol (compound **4**) and *cis*-phenyloxetan-3-ol (compound **6**), identified as novel PDE5 inhibitors. Their potency is reflected in lower IC_50_ values compared to icariin, the currently recognized potent natural inhibitor. This highlights their therapeutic potential.

Furthermore, while CO hydrosol exhibited only weak inhibitory effects against ACE, AChE, and ARG2, our findings suggest its potential in auxiliary roles, influencing erectile function through multiple physiological pathways. The *C. osmophloeum* hydrosol affects erectile dysfunction by inhibiting several enzyme’s activities. It inhibits AChE to enhance parasympathetic signaling, reduces ARG2 to preserve the eNOS co-substrate arginine, inhibits ACE to decrease the production of angiotensin, and most importantly, significantly inhibits PDE5 activity to promote smooth muscle relaxation (Figure 9). We also evaluated the cytotoxicity of CO hydrosol and its compounds on HepB3 (Appendix A), and the results indicated that both *trans*- and *cis*-phenyloxten-3-ol exhibited no cytotoxicity at 200 µM in Hep3B cells. This multifaceted impact underscores the prospects of CO hydrosol as a promising candidate for future development, possibly as a beverage or additive tailored to enhance erectile function.

Adding to this, recent research published in a leading pharmacology journal corroborates the potential of CO hydrosol in modulating erectile function. A study conducted by a team of researchers from a prominent university in Japan explored the effects of CO hydrosol on erectile dysfunction in animal models. Their findings suggest that the compounds present in CO hydrosol could enhance erectile function by improving blood flow to the penile tissues and reducing oxidative stress, providing further support for its potential as a natural remedy for erectile dysfunction.

## Figures and Tables

**Figure 1 plants-13-01518-f001:**
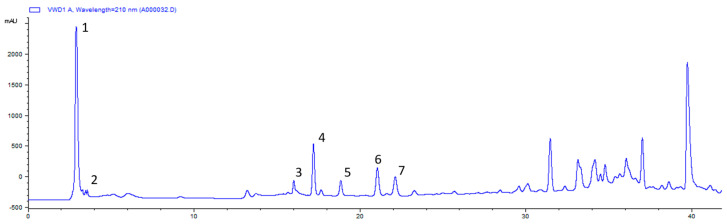
The profile spectrum of sub-fraction A of CO hydrosol water fraction. (The numbers 1–7 correspond to compounds **1**–**7**).

**Figure 2 plants-13-01518-f002:**
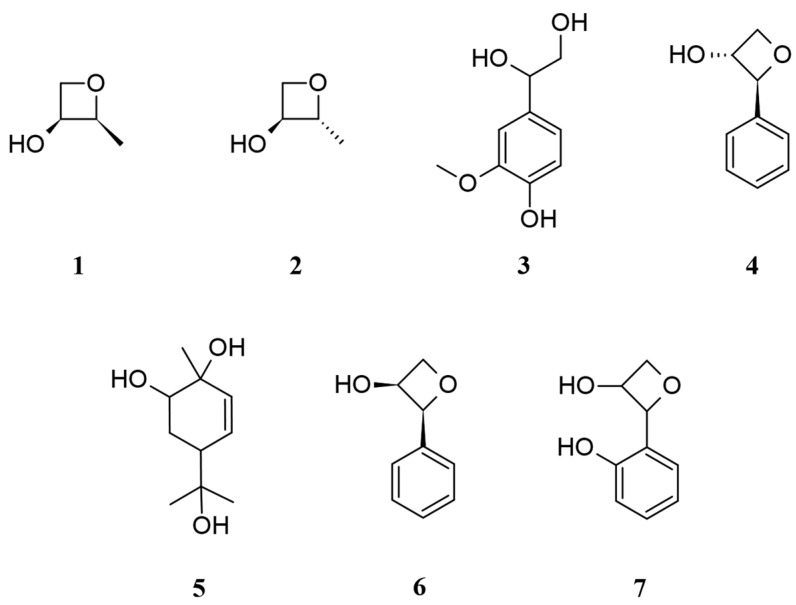
Structure of compounds **1** to **7**.

**Figure 3 plants-13-01518-f003:**
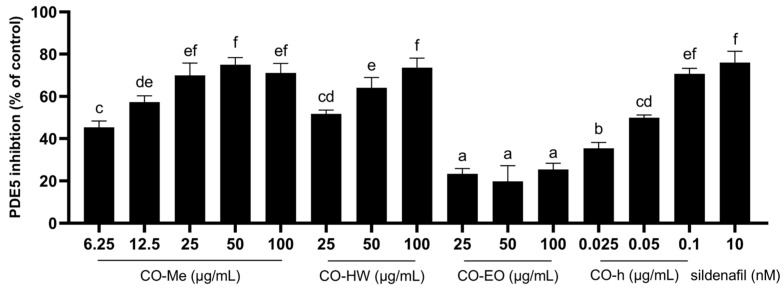
Inhibition activity of PDE5 by different CO leaf extracts (Me, MeOH extract; HW, hot water extract; EO, essential oil; h, hydrosol). Statistical analysis using one-way ANOVA. Different letters indicate significant differences between the different groups (*p* < 0.05).

**Figure 4 plants-13-01518-f004:**
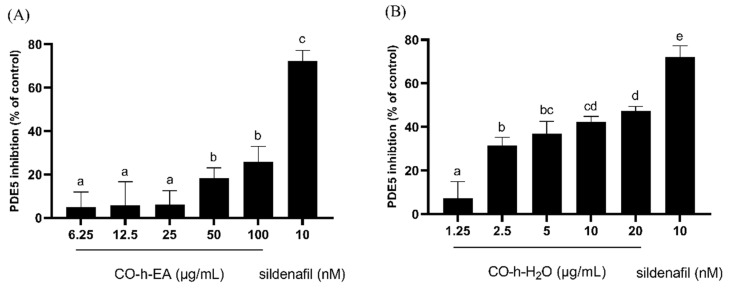
Inhibition activity of PDE5 by CO leaf hydrosol ((**A**), EA fraction; (**B**), water fraction). Statistical analysis using one-way ANOVA. Different letters indicate significant differences between the different groups (*p* < 0.05).

**Figure 5 plants-13-01518-f005:**
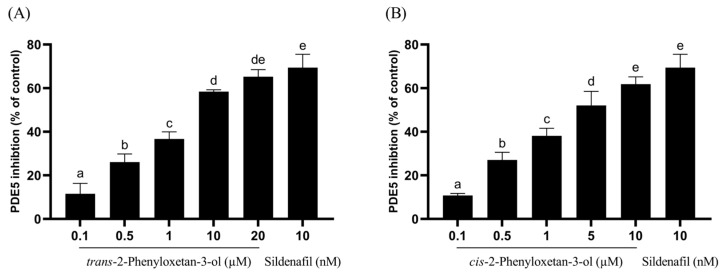
Inhibition activity of compounds isolated from CO leaf hydrosol ((**A**), *trans*-2-phenyloxetan-3-ol; (**B**), *cis*-2-phenyloxetan-3-ol). Statistical analysis using one-way ANOVA. Different letters indicate significant differences between the different groups (*p* < 0.05).

**Figure 6 plants-13-01518-f006:**
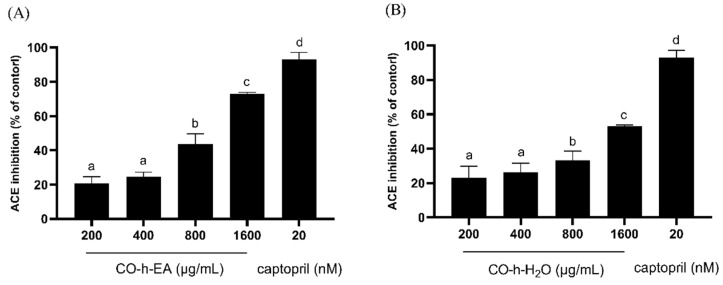
Inhibition activity of ACE by CO leaf hydrosol ((**A**), EA fraction; (**B**), water fraction). Statistical analysis using one-way ANOVA. Different letters indicate significant differences between the different groups (*p* < 0.05).

**Figure 7 plants-13-01518-f007:**
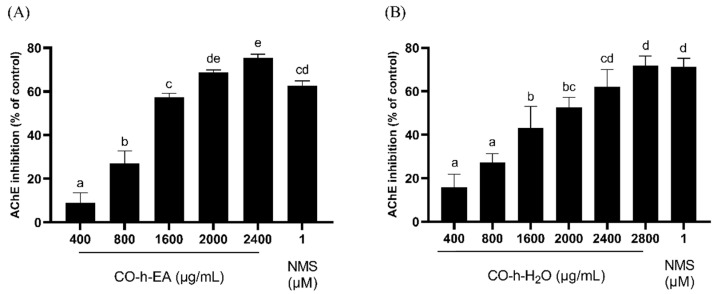
Inhibition activity of AChE by CO leaf hydrosol ((**A**), EA fraction; (**B**), water fraction). Statistical analysis using one-way ANOVA. Different letters indicate significant differences between the different groups (*p* < 0.05).

**Figure 8 plants-13-01518-f008:**
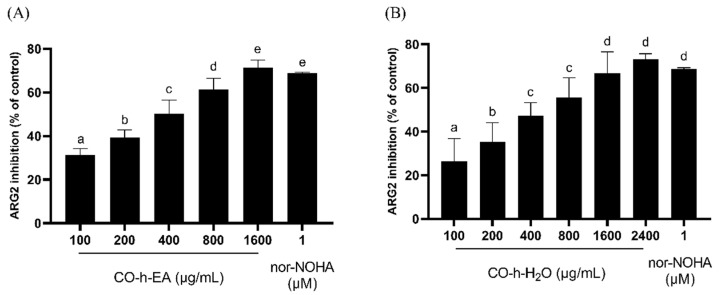
Inhibition activity of ARG2 by CO leaf hydrosol ((**A**), EA fraction; (**B**), water fraction). Statistical analysis using one-way ANOVA. Different letters indicate significant differences between the different groups (*p* < 0.05).

**Figure 9 plants-13-01518-f009:**
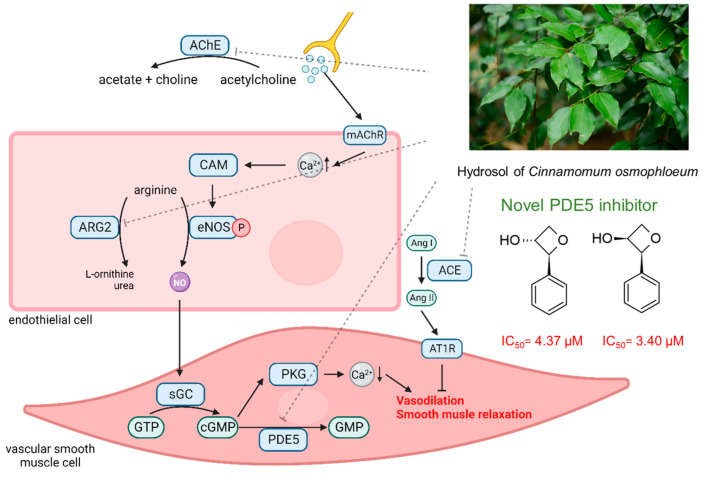
The mechanism by which CO hydrosol improves erectile dysfunction.

**Table 1 plants-13-01518-t001:** GC-MS analysis of CO essential oil and the hydrosol EA fraction.

RT (min)		Compound	Composition (%)		KI	Identification
EO	EA		EO	EA		
9.25	-	*α*-Pinene	1.25	-	936	KI, MS, ST
9.89	-	Camphene	0.70	-	952	KI, MS, ST
10.37	10.64	Benzaldehyde	0.41	0.66	965	KI, MS, ST
11.04	-	*β*-Pinene	0.32	-	979	KI, MS, ST
13.32	-	Limonene	0.17	-	1031	KI, MS, ST
-	14.13	Salicylaldehyde	-	0.21	1044	KI, MS
-	19.05	Vinylphenylcarbinol	-	0.35	1150	KI, MS
19.43	19.62	Benzenepropanal	0.93	0.72	1162	KI, MS
20.30	20.48	Terpinen-4-ol	0.03	0.02	1179	KI, MS, ST
20.99	21.18	*α*-Terpineol	0.01	0.11	1193	KI, MS, ST
21.12	21.3	Estragole	1.06	0.01	1196	KI, MS, ST
22.00	22.21	*cis*-Cinnamaldehyde	0.28	0.46	1216	KI, MS, ST
-	24.22	*p*-Allylphenol	-	0.34	1261	KI, MS
24.53	25.07	*trans*-Cinnamaldehyde	68.38	65.03	1279	KI, MS, ST
25.04	25.30	Bornyl acetate	1.03	0.15	1283	KI, MS, ST
-	26.25	Cinnamyl alcohol	-	1.94	1303	KI, MS, ST
28.00	28.17	Eugenol	0.76	0.78	1351	KI, MS, ST
29.01	-	*α*-Copaene	0.60	-	1374	KI, MS, ST
-	29.96	Vanillin	-	0.13	1389	KI, MS, ST
30.82	-	*β*-Caryophyllene	1.87	-	1416	KI, MS, ST
-	31.68	Coumarin	-	4.31	1430	KI, MS, ST
31.94	32.26	*trans*-Cinnamyl acetate	18.52	7.57	1444	KI, MS, ST
34.88	-	*δ*-Cadinene	1.00	-	1515	KI, MS
-	35.35	Ethyl 4-ethoxybenzoate	-	2.76	1519	KI, MS
37.30	-	Caryophyllene oxide	0.16	-	1577	KI, MS
48.50	-	Sclarene	0.36	-	1931	KI, MS

KI: Kovats retention index on DB-5MS column in reference to n-alkanes. ST: Authentic standard compounds. MS: NIST and Wiley library literature.

**Table 2 plants-13-01518-t002:** ^1^H and ^13^C NMR chemical shift data of compound **7** (δ ppm, in CD3OD).

Position	^13^C	^1^H	HMBC
2	69.1	4.47 (d, *J* = 4.4 Hz, 1H)	
3	69.3	3.88 (td, *J* = 4.4, 2.4 Hz, 1H)	
4	67.2	4.10 (ddd, *J* = 11.6, 4.8, 1.2 Hz, 1H)	C-3, C-6
		4.21 (dd, *J* = 11.6, 2.4 Hz, 1H)	
5	124.2	-	
6	155.5	-	
7	117.5	7.36 (dd, *J* = 7.6, 1.6 Hz, 1H)	C-5, C-9
8	130.3	7.18 (td, *J* = 8.4, 1.6 Hz, 1H)	C-6, C-10
9	121.8	6.93 (td, *J* = 7.6, 0.8 Hz, 1H)	C-5, C-7
10	132.0	6.81 (dd, *J* = 8.4, 0.8 Hz, 1H)	C-5, C-9

**Table 3 plants-13-01518-t003:** The mobile phase condition of RP-HPLC.

Retention (min)	Flow (mL/min)	MeOH (%)	H_2_O (%)
0	1.0	1	99
6	1.0	1	99
10	1.0	15	85
18	1.0	18	82
40	1.0	60	40
41	1.0	100	0
45	1.0	100	0
46	1.0	1	99
50	1.0	1	99

## Data Availability

All data generated or analyzed during this study are included in this published article and its Appendix A files.

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
