# Peer review of "Composition Characterization of Cinnamomum osmophloeum Kanehira Hydrosol and Its Enhanced Effects on Erectile Function"

_plants, 2024, doi:10.3390/plants13111518_

Round 1

Reviewer 1 Report

Comments and Suggestions for Authors

Two minor comments are explained or improved.

1. Please explaining why you use leaves instead of stem bark as plant materials.

2. Is Figure 9 a graphical abstract? If so, please redrawing it based on the reliable results of this study. If not, please providing a detailed explanation of its specific meaning and mark it in the text.

Author Response

Reviewer 1

Commend 1. Please explaining why you use leaves instead of stem bark as plant materials.

Response:

We are very grateful for your comments on the manuscript. Cinnamomum osmophloeum is an endemic plant in Taiwan, and we are particularly interested in the bioactivity and components of its leaves. In Taiwan, the leaves of C. osmophloeum are used to extract essential oil, producing a large amount of hydrosol as a byproduct. This hydrosol has been widely used, even as a beverage. However, no studies have investigated the components and biological activities of C. osmophloeum leaf hydrosol. Therefore, we have focused our research on the leaves rather than the stem bark in this study.

Commend 2. Is Figure. 9 a graphical abstract? If so, please redrawing it based on the reliable results of this study. If not, please providing a detailed explanation of its specific meaning and mark it in the text.

Response:

Thank you for your thoughtful suggestion. We have modified the graphic and added a text explanation of Figure 9 in lines 386-390. We described how the hydrosol affects these enzymes to improve erectile dysfunction and its probable mechanism. Thank you for bringing this to our attention.

Reviewer 2 Report

Comments and Suggestions for Authors

The manuscript aims at investigating the Cinnamomum osmophloeum hydrosol as an enhancer of the erectile function. However, some concerns arise.

l. 55. The sentence should be rephrased (polar compounds, not aqueous compounds).

l. 107. Regarding the identification of the compounds 1-6, references should be embedded, accordingly.

Due to the coumarin hepatotoxicity, bring light into the safety of the hydrosol. The data on the toxicity of phenyloxetan 3-ol and the main compounds should be also provided.

With respect to the enzyme inhibitory activity, the assays were performed with different concentration ranges (see Fig. 4, 5-8) reaching 2400/2800 μg/ml. No problem with the solubility?

Starting point in the study is the inhibition towards PDE5 by the aqueous fraction in concentrations up tom 20 μg/ml. Thus, for the other enzymes, the inhibitory activity is week, not moderate.

Provide the voucher specimen number in Plant material subsection.

l.249 Please, rephrase the sentences 15 year old tree?

Concerning the hydrosol yield, it is impressive that 30 kg CO leaves (from 15 years old tree) afforded 0.13% ethyl acetate fraction and 0.004% aqueous fraction. Is it efficient to explore the endemic species in this way?

l. 375. The last paragraph should be moved in the Discussion section together with appropriate reference.

Please, check for typos errors.

Author Response

Reviewer 2

Commend 1. l. 55. The sentence should be rephrased (polar compounds, not aqueous compounds).

Response:

We sincerely apologize for the diction error. We have corrected it in the revised version. Thank you for bringing it to our attention.

Commend 2. l. 107. Regarding the identification of the compounds 1-6, references should be embedded, accordingly.

Response:

Thank you for your thoughtful suggestion. We have embedded the reference of compound 1-6 from Lines108 to 111.

Commend 3. Due to the coumarin hepatotoxicity, bring light into the safety of the hydrosol. The data on the toxicity of phenyloxetan-3-ol and the main compounds should be also provided.

Response:

We appreciate your suggestion. Due to the hepatotoxicity of coumarin, we confirmed the cytotoxicity of the hydrosol and its main compounds using the Hep3B cell line. The results are presented in Supplementary Information S20. The results indicate that both trans- and cis-phenyloxten-3-ol exhibit no cytotoxicity at 200 µM in Hep3B cells (lines 390- 393).

Commend 4. With respect to the enzyme inhibitory activity, the assays were performed with different concentration ranges (see Fig. 4, 5-8) reaching 2400/2800 μg/ml. No problem with the solubility?

Response:

Both the ethyl acetate fraction and the water fraction were dissolved in DMSO. During the enzyme inhibitory activity assay, even at high concentrations, the solution did not exhibit noticeable turbidity or the formation of insoluble substances after being mixed with the reaction buffer.

Commend 5. Starting point in the study is the inhibition towards PDE5 by the aqueous fraction in concentrations up tom 20 μg/ml. Thus, for the other enzymes, the inhibitory activity is weak, not moderate.

Response:

In our study, we focused on the PDE-5 inhibitory activity because the hydrosol showed a strong effect. We identified the active compounds as phenyloxetan-3-ol. Since the PDE-5 enzyme is highly related to erectile dysfunction, we also measured the activity of other enzymes related to ED. Although the results showed a weak effect on these enzymes, we believe that even a weak effect can be beneficial. We value your suggestion and have changed the word in the conclusion from "modest" to "weak" (line 384).

Commend 6. Provide the voucher specimen number in Plant material subsection.

Response:

We have provided the voucher specimen information in line 253. Thank you for your suggestion.

Commend 7. l.249 Please, rephrase the sentences 15 year old tree?

Response:

We appreciate your suggestion. In the revised version, we have adjusted the sentence to: “The leaves used for essential oil and hydrosol preparation were collected from a batch of 15-year-old C. osmophloeum trees” (lines 258-259).

Commend 8. Concerning the hydrosol yield, it is impressive that 30 kg CO leaves (from 15 years old tree) afforded 0.13% ethyl acetate fraction and 0.004% aqueous fraction. Is it efficient to explore the endemic species in this way?

Response:

As pointed out by the reviewer, the hydrosol preparation method extracts only volatile compounds from the plant materials. Given the low yield of essential oil, it is inevitable to obtain a lower yield of chemicals in the hydrosol. Despite this, we believe that identifying the high-polarity compounds in the hydrosol, while not highly efficient, is very unique. The hydrosol market is growing, but it lacks precise chemical analysis. Therefore, we believe that identifying and exploring the biological activities of these trace components is a highly valuable part of our research, contributing significantly to the understanding of hydrosols.

Commend 9. l. 375. The last paragraph should be moved in the Discussion section together with appropriate reference.

Response:

Thank you for your thoughtful suggestion. We have modified the graphic and added a text explanation of Figure 9 in lines 386-390. We described how the hydrosol affects these enzymes to improve erectile dysfunction and its probable mechanism. Thank you for bringing this to our attention.

Round 2

Reviewer 2 Report

Comments and Suggestions for Authors

Authors addressed all my comments.